behaviour/ecology/health and disease and epidemiology

diet, foraging, haematology, heterophil : lymphocyte ratio, northern gannets

**Author for correspondence:**
M. Jessopp
e-mail: m.jessopp@ucc.ie

# Evidence of links between haematological condition and foraging behaviour in northern gannets (*Morus bassanus*)

Z. Malvat[1], S. A. Lynch[1,3], A. Bennison[1,2] and M. Jessopp[1,2]

[1]School of Biological, Earth and Environmental Sciences, [2]MaREI Centre, Environmental Research Institute, and [3]Aquaculture and Fisheries Development Centre and Environmental Research Institute, University College Cork, Cork, Ireland

SAL, 0000-0002-4542-7150; AB, 0000-0001-9713-8310; MJ, 0000-0002-2692-3730

Haematological analyses can reveal the physiological condition of birds, which are known to efficiently disguise symptoms of stress and disease. However, the interpretation of such analyses requires species-specific baseline data, which are lacking for most free-living seabird species. We provide baseline reference data for several haematological parameters in northern gannets (*Morus bassanus*) and combine this with telemetry and dietary data to understand the links between haematological condition and foraging behaviour. Blood samples were collected from breeding northern gannets in July 2017 (*n* = 15) and 2018 (*n* = 28), which were also equipped with GPS tags. Smears were prepared for performing blood cell counts, including immature erythrocyte and microcyte percentages, total and differential leucocyte counts, heterophil : lymphocyte (H : L) ratio and total thrombocyte count; the remaining blood was used for stable isotope analysis and foraging behaviours were inferred from the recovered tag data. Blood cell counts revealed that the sampled birds were highly stressed and some showed an immune response, evident from the abnormal leucocyte counts and the H : L ratio. There were no sex-related differences in haematological parameters or diet, in contrast to foraging parameters where females undertook longer trips than males and spent proportionately more time in search behaviours. The percentage time spent actively foraging was weakly negatively correlated with the percentage of eosinophils. While there was no direct link between haematological condition and diet, one bird feeding at a

relatively low trophic level undertook exceptionally short foraging trips and showed abnormal blood cell counts. This suggests a link between haematological condition and foraging ecology that can be employed in assessing seabird health.

## 1. Introduction

Haematology has been described as an indispensable tool for the assessment of health and disease in birds by several authors [1–4]. The avian haemogram, i.e. complete blood cell count, is generally evaluated by counting the different blood cell types per microlitre of blood and evaluating their quality. Several techniques, both manual and automated, are available for performing blood cell counts [2,5–7], but many of them cannot be employed in the field due to technical requirements and logistical constraints. In such situations, a well-prepared blood smear can be used to evaluate various haematological indices. In addition to performing blood cell counts, smears can also be used to assess the presence of parasites and the associated parasitaemia [2,3,8]. The major cellular components of avian blood include erythrocytes (red blood cells or RBC), leucocytes (white blood cells or WBC), which are further classified as granulocytes, including heterophils, eosinophils and basophils, and agranulocytes (or mononuclear cells), including monocytes and lymphocytes—and thrombocytes, all of which are nucleated in birds [5]. To facilitate the evaluation of the avian haemogram, packed cell volume (PCV), total erythrocyte count, haemoglobin concentration, total leucocyte and thrombocyte counts, and leucocyte differentials can be used [4], though it must be noted that not all of these can be obtained using only a blood smear. These haematological analyses allow the measurement of quantitative and qualitative changes in the blood cell fractions and their morphology, which can reveal information about the physiological state of the individual ([9] as read in [10]).

A vast majority of haematological abnormalities are quantitative in nature, characterized by either an excessive or deficient production of one or more cellular components of blood [11]. A decrease in the PCV or total erythrocyte count is indicative of anaemia, whereas elevated values indicate polycythaemia. In addition, anaemia can also be indicated by the presence of polychromatophilic or immature erythrocytes (polychromasia) and/or malformed or small erythrocytes (microcytes) in numbers greater than 5% [12]. Anaemia can result from acute blood loss or haemorrhage, destruction of erythrocytes such as by blood parasites or toxicosis, or due to a decreased production of erythrocytes in the bone marrow [4]. Any form of anaemia results in a reduced oxygen-carrying capacity of the affected cells, which subsequently leads to an overall reduction in the fitness of the individual [5,12,13]. Polycythaemia may occur as a result of dehydration, but it is rarely reported in birds [3,5]. Some haematological abnormalities can also be qualitative, such as hypochromasia and sickle cell anaemia. Hypochromasia is characterized by pale-staining erythrocytes which are deficient in haemoglobin, and it may be indicative of iron deficiency or inflammation [14]. Leucocytosis, or an increase in levels of leucocytes, may result from an infection, inflammation, trauma or as a response to a toxicity. The heterophil : lymphocyte (H : L) ratio is being increasingly used to assess stress associated with starvation, temperature extremes, infection or psychological disturbance in birds [15–18]. An elevated H : L ratio is an effective indicator of chronic stress in birds, both captive and free-living, as opposed to the more conventional corticosterone levels, which can only be used to assess acute stress [19]. A reduction in thrombocyte levels, also known as thrombocytopenia, may occur in response to septicaemia, whereas thrombocytosis accompanied by an increase in the size of thrombocytes may indicate chronic inflammation in birds [20].

An accurate interpretation of the avian haemogram requires knowledge of the haematological characteristics of birds in healthy condition, and the changes these characteristics show in response to stress or disease. Furthermore, it becomes important to distinguish the sources of natural variations, such as differences due to sex, diurnal and seasonal variations, from the effects of environmental or physiological stress [16,21]. However, such baseline information is largely undocumented for a large number of avian species, and most of the information available for avian haematological parameters has been extrapolated from values established for parrots (psittacines) [7,18,21–23]. Seabirds are considered to be effective monitors of the health of aquatic systems and are often employed as proxies to assess the effects of stressors, both natural and anthropogenic, on the aquatic environment [24]. Since seabirds forage over large areas and many undertake long migrations, they have the potential to introduce pathogens to susceptible communities that may be lacking immunity [25]. Seabirds are known to show both dramatic, e.g. seabird die-offs, and sub-lethal physiological responses to environmental change, and the responses of the latter type can be used as indicators of ecosystem changes. However, birds are

known to disguise symptoms but not exhibit any clinical signs until very late in the disease process [21]. In such cases, haematological analyses become very important, as they can be used to identify subtle physiological effects. In spite of this fact, there remains a large paucity with regard to species-specific haematological indices, as they remain largely unstudied or unreported [22,23].

Furthermore, haematological analyses have seldom been employed in conjunction with telemetry and dietary analyses to understand seabird behaviour (see [26,27]). The northern gannet *Morus bassanus*, hereafter gannet, is the largest seabird breeding in the North Atlantic, with UK and Ireland supporting 68% of the world population [28]. Gannets are important marine predators, consuming a variety of pelagic fish [29,30] and also taking discards from fishing trawlers [31–33]. While sex-specific differences [34] and individual variability [32] in foraging behaviour and utilization of fishery discards have been noted, a negative correlation between the proportion of discards in the diet and individual body condition [32] suggests that either discards are poor quality food or that poor quality individuals consume discards. However, a link between haematological condition, diet and foraging behaviour is yet to be established. This study therefore aimed to (i) determine the physiological condition of breeding gannets on the Great Saltee island for the first time using haematological analysis to assess individual variability and sex-specific differences, (ii) explore the relationships between various haematological parameters, and (iii) explore potential links between body condition, foraging behaviour and diet. It is expected that the results will provide evidence for the usability of haematological analyses in understanding seabird foraging ecology.

# 2. Material and methods

Blood and feather samples were collected from breeding gannets attending two- to four-week-old chicks on Great Saltee, Co. Wexford, Ireland, in July 2017 ($n = 15$) and 2018 ($n = 28$). This is the third largest gannet colony in Ireland, supporting approximately 4722 breeding pairs [35]. Birds were captured using a metal crook at the end of a 4–6 m pole during change-over, targeting the bird about to leave the nest. A small amount (less than 2 ml, approx. 0.07% volume/weight) of blood was drawn from the tarsal vein of each bird before being fitted with a leg ring with a unique identifier, and two to three breast feathers were plucked for genetic sexing following Griffiths *et al.* [36]. The birds were also tagged with a GPS tracker (i-gotu GT120, mobile action technology, 17 g) set to record location at 3 min intervals. Tags were attached to two to three central tail feathers using three to four strips of a 15 mm wide waterproof Tesa© tape. Total deployment weight was less than 2% of adult body weight.

The preparation of blood smears differed slightly between sampling years. Because blood smears could not be processed immediately for cell counting in the field, blood was mixed with ethylenediaminetetraacetic acid (EDTA), a commonly used anticoagulant [37]. In 2017, a small amount of blood was mixed with a variable amount of EDTA to determine the smallest amount needed to prevent clot formation, which resulted in over- or underdilution of some samples. In 2018, 60 µl of blood was mixed with an equal volume of EDTA, with a standardized 60 µl of this mixture used for preparing smears.

Blood smears were prepared using the two-slide wedge technique [38], which were then allowed to air dry. Blood smears from July 2017 were stained in February 2018, and those from July 2018 were stained in April 2019, using One Step Wright's Stain (Sigma-Aldrich). The air-dried slides were fixed in methanol for 2 min, then dipped in One Step Wright's Stain for 15–30 s, followed by dipping in rinse solution no. 2 (Sigma-Aldrich) for approximately 45 s and finally dipped in deionized distilled water for 25 s using quick dips. The slides were then allowed to air dry for a few minutes, mounted in Eukitt® quick-hardening mounting medium (Sigma-Aldrich), coverslipped and allowed to stand for 5 days for the mounting medium to dry. These slides were then examined under a microscope at 40–100× magnification, and cell counts were performed. Blood cells were counted and staged (in the case of erythrocytes) based on Campbell [5,37]. To ascertain the levels of anaemia, the number of immature erythrocytes, microcytes, hypochromic erythrocytes and poikilocytes were counted per 100 erythrocytes at 40× magnification, performing three replicates on each slide. The presence of immature erythrocytes and microcytes in numbers greater than 6.78% and 5% was considered to be indicative of regenerative and microcytic anaemia, respectively [12]. To estimate the total WBC count, leucocytes were counted per field at 40× magnification for 10 consecutive fields. The total number of leucocytes counted was divided by 10 (i.e. averaged) and then multiplied by 2000 to derive the total WBC count [39,40]. A differential WBC count was performed by counting the different types of leucocytes, i.e. heterophils, eosinophils, basophils, lymphocytes and monocytes at 100× magnification with oil immersion till the total reached 100. The percentage of each type of leucocyte was then determined to obtain the WBC differential [2]. To obtain the

heterophil : lymphocyte (H : L) ratio, only heterophils and lymphocytes were counted at 100× magnification with oil immersion until the total (heterophil + lymphocyte) reached 100. The H : L ratio was then calculated by dividing the number of heterophils by the number of lymphocytes [17]. To minimize errors due to uneven distribution of leucocytes in the smears, all counts were performed following the modified battlement technique described by Bain [41], and smear cells (disrupted or smudged cells) that could be identified were included in the counts. The thrombocyte count was obtained by counting the total number of thrombocytes per 1000 erythrocytes at 100× magnification with oil immersion. Reference values for total and differential WBC counts and thrombocyte counts were calculated as mean ± 1 s.d. (68% confidence interval). For the H : L ratio, only the mean + 1 s.d. was considered, since lower H : L ratios do not indicate any abnormality. For smears obtained in July 2017, only the erythrocyte and thrombocyte counts could be performed, as the deterioration of granules within heterophils and eosinophils between staining and identification meant that cell types could no longer be distinguished. Similarly, total WBC counts could not be performed on 2017 smears due to the inconsistent dilutions of EDTA used.

Gannet foraging behaviours were inferred from the recorded GPS tracks for the birds whose tags were recovered using hidden Markov models in the *moveHMM* package in R (v. 3.5.2, [42]). The proportion of time spent resting, travelling and foraging/searching during each trip [43], the number of trips undertaken, furthest distance travelled from the colony, longest trip undertaken and the average trip length were extracted for each bird.

Remaining blood samples were stored in a flask on ice for up to a maximum of 8 h before being centrifuged at 1600 r.p.m. for 10 min to separate the blood cells and plasma. The samples were then dried at 60°C before being ground to a fine powder. Lipids were removed from the ground samples by the repeated addition of 2 : 1 chloroform : methanol mixture, with the supernatant containing dissolved lipids being siphoned off using a pipette. These samples were then sent for the analysis of carbon and nitrogen stable isotopes at Elemtex Laboratories, UK.

Haematological, behavioural and stable isotope data were analysed using R (v. 3.5.2, [42]). Haematological and behavioural variables were tested for normality, and differences between sex and year were tested using either the parametric independent samples *t*-test or the non-parametric Mann–Whitney *U*-test, accordingly. Pearson's product–moment correlation and Spearman's correlation tests were used to determine relationships between haematological and behavioural variables.

## 3. Results

### 3.1. Bird characteristics

Of the 45 birds sampled across both years, 21 (47%) were male, and 22 (49%) were female, with the sex of two birds being undetermined. While females were on average heavier than males in both years (2017, male average = 2.82 ± 0.27 kg, female average = 3.02 ± 0.27 kg; 2018, male average = 2.86 ± 0.49 kg, female average = 3.03 kg), the difference was not statistically significant (Mann–Whitney *U*-test and *t*-test, $p > 0.05$; tables 6 and 7), and the weights of all birds fell within the normal range described for gannets (table 1).

### 3.2. Haematological analyses

Since the majority of cell counts could not be performed on smears from 2017 due to intracellular granule disintegration, results pertain to the 2018 smears, unless stated otherwise. Mature as well as immature erythrocytes, heterophils, lymphocytes, eosinophils and thrombocytes were present in all smears, while microcytes and monocytes were observed only in some smears (figure 1). No basophils were encountered in any of the smears. Poikilocytes and hypochromic erythrocytes were only rarely observed (figure 1). The stages of immature erythrocytes included the basophilic rubricyte, early polychromatic rubricyte, late polychromatic rubricyte and polychromatic erythrocyte (figure 1). Of the leucocytes, heterophils were the most abundant, followed by eosinophils, lymphocytes and monocytes in decreasing order. Mature erythrocytes were the largest cells to be observed, followed by leucocytes and then by thrombocytes. Among the leucocytes, eosinophils and monocytes were similar in size to heterophils, while lymphocytes showed much variation in size.

Blood cell counts were available for 27 birds from 2018, of which 12 birds (44%) had all their counts within the reference range (table 1), while other birds had values outside 1 s.d. of the mean, for one or more haematological parameters (table 2). For 2017, cell counts were available for nine birds, of which four (44%) birds had all the counts within the reference range.

**Table 1.** Reference ranges for weight and haematological parameters of northern gannets taken from the literature and this study. Reference ranges from this study are mean ± 1 s.d. using 2018 samples.

| parameter | reference range |
|---|---|
| weight (kg) | 2.3–3.6[a] |
| immature erythrocyte % | 0–6.78[b] |
| microcyte % | 0–5[b] |
| total WBC count (cells per µl) | 2953–6677 |
| H : L ratio | 5.85 |
| heterophil % | 50–66 |
| lymphocyte % | 11–21 |
| eosinophil % | 17–33 |
| monocyte % | 0–2 |
| thrombocyte count (cells per 1000 erythrocytes) | 5–36 |

[a][44].
[b][12].

Gannet D37 (male) showed the highest total WBC and monocyte count, while D25 had the lowest WBC count; both values were outside the calculated reference range. D20 and D25 (females), and D26 (male) had immature erythrocyte values higher than 6.78%. The highest H : L ratio observed was 9.1 in D27 (male). This particular bird had the lowest observed values for lymphocytes and thrombocytes, and the highest observed % value for eosinophils; however, its other cell counts were well within the 'normal' range. No birds showed 'abnormal' values (values outside the reference range) for more than four of the nine haematological parameters simultaneously.

There was considerable individual variation in the different blood cell counts, but no significant differences between sexes for any parameter (all $p > 0.05$; tables 3, 6 and 8). Two blood parameters, the microcyte % in males (Mann–Whitney $U$-test, $W = 38.5$, $p = 0.012$) and thrombocyte count in females (Mann–Whitney $U$-test, $W = 11$, $p = 0.038$), differed significantly between 2017 and 2018 (table 3).

## 3.3. Stable isotope analysis and foraging trip characteristics

No significant differences occurred in the carbon and nitrogen stable isotope values between the two sexes (table 4), although $\delta^{15}N$ values were significantly higher in 2017 than in 2018, for both sexes (tables 9 and 10). Some birds that exhibited 'abnormal' values for multiple haematological parameters (e.g. D26, D27, D40) in 2018 were feeding at relatively low trophic levels (figure 2), indicated by comparatively lower $\delta^{15}N$ values, but this was not consistent, with D25 feeding at a comparatively high trophic level. Furthermore, birds that appeared to be feeding at relatively very low trophic levels, such as D46, had most of their counts within the 'normal' range.

Tracking data were available for 16 birds (eight males and eight females) in 2017 and 21 birds (12 males and nine females) in 2018. While in 2017 there were no sex-based differences in foraging parameters (table 7), in 2018 females undertook significantly longer trips than males ($t = 1.82$, d.f. = 18.79, $p = 0.046$) and had a slight tendency to travel further from the colony, but this was not significantly different from males ($t = 1.69$, d.f. = 18, $p > 0.05$). However, females also spent a significantly lower percentage of time searching for food ($t = -3.83$, d.f. = 18.95, $p = 0.0011$), and a greater percentage of time resting ($t = 2.82$, d.f. = 18, $p = 0.012$) than males, whereas the percentage of time spent travelling was similar in both sexes ($t = 0.26$, d.f. = 15.14, $p > 0.05$) (table 5).

In 2018, the percentage time spent searching was weakly negatively correlated with the eosinophil percentage (Pearson's product–moment correlation, $r = -0.54$, d.f. = 18, $p = 0.014$), i.e. there was a decrease in percentage time spent searching as the percentage of eosinophils increased. The total WBC count also showed a weak positive correlation with the thrombocyte count (Spearman's rank correlation, $r_s = 0.51$, $n = 26$, $p < 0.001$), i.e. birds with higher WBC counts had higher counts for thrombocytes. No significant correlations were observed between any of the remaining haematological and behavioural parameters (tables 11 and 12).

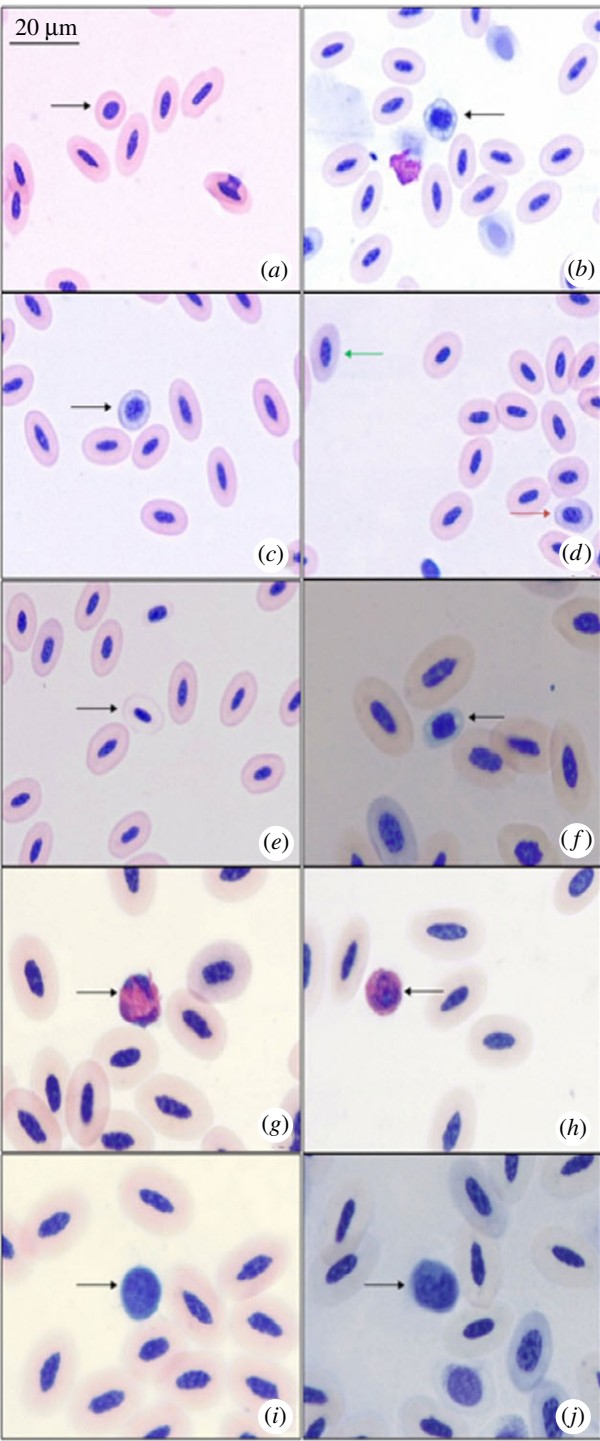

**Figure 1.** Various blood cells observed in gannet blood smears: (*a*) microcyte (black arrow) surrounded by mature erythrocytes, (*b*) basophilic rubricyte, (*c*) early polychromatic rubricyte, (*d*) late polychromatic rubricyte (orange arrow) and polychromatic erythrocyte (green arrow), (*e*) hypochromic erythrocyte, (*f*) thrombocyte, (*g*) heterophil, (*h*) eosinophil, (*i*) lymphocyte and (*j*) monocyte. Images (*a*)–(*e*) under 40× magnification, images (*f*)–(*j*) under 100× magnification with oil immersion.

## 4. Discussion

For the first time, this study provides haematological reference values for breeding gannets and shows a potential link between haematological parameters and foraging behaviour. Gannet leucocytes were visibly smaller than erythrocytes, in contrast to most avian species where heterophils, monocytes and eosinophils are larger than erythrocytes [10,45]. However, it was not possible to determine whether

**Table 2.** Blood cell counts for northern gannet blood samples from the Great Saltee, collected in July 2017 (D01–D20) and 2018 (D25–S010). Values outside the reference range have been highlighted—higher than normal in orange, lower than normal in blue. F, female; M, male; U, sex not known; I.E., immature erythrocyte; Thr, thrombocyte count per 1000 erythrocytes.

| bird id. | sex | weight (kg) | I.E. (%) | microcyte (%) | total WBC | H : L ratio | heterophil (%) | lymphocyte (%) | eosinophil (%) | monocyte (%) | Thr |
|---|---|---|---|---|---|---|---|---|---|---|---|
| D01 | M | 3.05 | 6.67 | 6.33 | — | — | — | — | — | — | 2 |
| D02 | M | 2.8 | 3.76 | 2 | — | — | — | — | — | — | 1 |
| D03 | U | 2.8 | 4.67 | 0 | — | — | — | — | — | — | 37 |
| D04 | F | 2.8 | 6 | 1.33 | — | — | — | — | — | — | 17 |
| D05 | F | 2.9 | 2.67 | 2.67 | — | — | — | — | — | — | 1 |
| D08 | M | 2.5 | 6 | 2.33 | — | — | — | — | — | — | 28 |
| D14 | F | 3.1 | 6.67 | 3.67 | — | — | — | — | — | — | 2 |
| D18 | F | 3.15 | 6.67 | 0 | — | — | — | — | — | — | 14 |
| D20 | F | 2.6 | 8 | 2 | — | — | — | — | — | — | 0 |
| D25 | F | 3.25 | 7.33 | 0 | 2200 | 5.67 | 68 | 11 | 21 | 0 | 4 |
| D26 | M | 2.8 | 8.67 | 0.33 | 3800 | 2.96 | 57 | 19 | 23 | 1 | 1 |
| D27 | M | 3.45 | 4.67 | 1.33 | 3200 | 9.1 | 54 | 6 | 40 | 0 | 3 |
| D28 | M | 2.6 | 3.33 | 0.67 | 5800 | 4.56 | 65 | 14 | 21 | 0 | 23 |
| D30 | F | 2.75 | 6.67 | 0.33 | 5600 | 3.34 | 48 | 17 | 35 | 0 | 34 |
| D31 | M | 3.5 | 5.33 | 0 | 6800 | 2.7 | 58 | 18 | 23 | 1 | 27 |
| D32 | F | 3.05 | 5 | 0 | 7800 | 2 | 43 | 25 | 32 | 0 | 17 |
| D33 | F | 3.25 | 4.33 | 0 | 5200 | 7.33 | 60 | 10 | 28 | 2 | 34 |
| D34 | M | 2.7 | 5.33 | 0.33 | 5000 | 4.94 | 65 | 12 | 23 | 0 | 32 |
| D35 | F | 2.8 | 4 | 1 | 4600 | 3.81 | 68 | 16 | 16 | 0 | 3 |
| D36 | M | 2.8 | 4 | 0 | 4400 | 3.3 | 57 | 16 | 27 | 0 | 10 |
| D37 | M | 2.75 | 5 | 1.33 | 11 000 | 3.21 | 53 | 17 | 25 | 5 | 29 |
| D38 | F | 3.15 | 4.67 | 0 | 2800 | 2.33 | 54 | 22 | 22 | 2 | 12 |

(Continued.)

**Table 2.** (*Continued.*)

| bird id. | sex | weight (kg) | I.E. (%) | microcyte (%) | total WBC | H : L ratio | heterophil (%) | lymphocyte (%) | eosinophil (%) | monocyte (%) | Thr |
|---|---|---|---|---|---|---|---|---|---|---|---|
| D40 | F | 3 | 6.33 | 0.67 | 2400 | 2.33 | 50 | 23 | 21 | 6 | 4 |
| D41 | M | — | 4.67 | 1.67 | 3200 | 3.25 | 46 | 14 | 37 | 3 | 44 |
| D42 | F | 2.8 | 5.67 | 0.67 | 3800 | 2.22 | 45 | 20 | 34 | 1 | 17 |
| D43 | F | 3.45 | 6 | 0.67 | 5200 | 4.26 | 56 | 11 | 33 | 0 | 32 |
| D44 | M | 3.2 | 3.33 | 2 | 5400 | 2.84 | 58 | 23 | 19 | 0 | 76 |
| D45 | M | 2.65 | 3.67 | 1.67 | 4000 | 3.54 | 58 | 17 | 23 | 2 | 12 |
| D46 | F | 3.2 | 3 | 6 | 3200 | 3 | 57 | 19 | 23 | 1 | 19 |
| D47 | M | 2.85 | 6.33 | 0 | 4800 | 3.76 | 61 | 13 | 26 | 0 | 16 |
| D48 | F | 2.9 | 5 | 0.67 | 3800 | 3.35 | 49 | 14 | 36 | 1 | 19 |
| D50 | M | 3.35 | 6.33 | 0.33 | 4200 | 3.35 | 64 | 22 | 11 | 3 | 16 |
| D51 | M | 3.05 | 5.33 | 1 | 5000 | 6.69 | 66 | 12 | 21 | 1 | 13 |
| D52 | F | 2.95 | 5.67 | 1 | 4600 | 5.25 | 79 | 13 | 8 | 0 | 12 |
| D53 | M | 2.8 | — | — | — | — | — | — | — | — | — |
| D54 | U | 3.4 | 4.67 | 1.33 | 4600 | 3.54 | 66 | 21 | 13 | 0 | 17 |
| S010 | F | 2.9 | 5.67 | 0 | 7600 | 7.33 | 62 | 9 | 29 | 0 | 32 |

**Table 3.** Average values of various haematological parameters for male and female northern gannets on Great Saltee, Ireland. Values indicate mean ± s.d. For heterophils, lymphocytes, eosinophils and monocytes, values outside the bracket indicate % proportion and values inside the bracket indicate the absolute counts (calculated from the % proportion and the total WBC counts for individual birds and then averaged). Values with the same superscript letter in a column are significantly different (detailed statistics provided in the appendix).

| parameter | year | female (mean ± s.d.) | male (mean ± s.d.) |
|---|---|---|---|
| immature erythrocyte % | 2017 | 6.00 ± 2.00 | 5.45 ± 1.57 |
| | 2018 | 5.33 ± 1.17 | 5.07 ± 1.47 |
| microcyte % | 2017 | 1.93 ± 1.38 | 3.55 ± 2.41[a] |
| | 2018 | 0.85 ± 1.6 | 0.82 ± 0.72[a] |
| total WBC count (cells per µl) | 2018 | 4523.08 ± 1778.65 | 5123.08 ± 2037.22 |
| H : L ratio | 2018 | 4.02 ± 1.85 | 4.17 ± 1.84 |
| heterophil % (cells per µl) | 2018 | 56.85 ± 10.42 | 58.62 ± 5.69 |
| | | (2550.15 ± 1041.7) | (3002.77 ± 1122.43) |
| lymphocyte % (cells per µl) | 2018 | 16.15 ± 5.32 | 15.62 ± 4.54 |
| | | (717.08 ± 404.35) | (818.62 ± 424.78) |
| eosinophil % (cells per µl) | 2018 | 26.00 ± 8.38 | 24.54 ± 7.37 |
| | | (1224.15 ± 714.09) | (1224.15 ± 526.10) |
| monocyte % (cells per µl) | 2018 | 1.00 ± 1.68 | 1.23 ± 1.59 |
| | | (31.69 ± 46.14) | (77.54 ± 147.15) |
| thrombocyte count (cells per 1000 erythrocytes) | 2017 | 6.8 ± 8.04[b] | 10.33 ± 15.31 |
| | 2018 | 18.38 ± 11.53[b] | 23.23 ± 19.90 |

**Table 4.** Results from stable isotope analysis from 2017 and 2018. Values with the same superscript letter in a column are significantly different (detailed statistics provided in tables 6 and 9). Note that *p*-values are for sex-based differences and not for year-based differences.

| parameter | year | female (mean ± s.d.) | male (mean ± s.d.) | test statistic | *p*-values |
|---|---|---|---|---|---|
| $\delta^{15}N$ | 2017 | 14.77 ± 0.55[a] | 14.76 ± 0.22[b] | 0.04 | 0.969 |
| | 2018 | 13.34 ± 0.47[a] | 13.75 ± 0.69[b] | −1.75 | 0.095 |
| $\delta^{13}C$ | 2017 | −18.27 ± 0.35 | −18.20 ± 0.29 | −0.28 | 0.792 |
| | 2018 | −18.68 ± 0.35 | −18.48 ± 0.33 | −1.52 | 0.142 |

this is the normal condition for gannets, or if this was the result of the season or breeding status. All sampled birds were heterophilic, i.e. heterophils were the most abundant WBC type, followed by eosinophils and then lymphocytes. While this is generally consistent with most birds where heterophils and lymphocytes are the most abundant types of circulating leucocytes [18], deviations from this generalization have previously been reported [46,47]. Since baseline values for the differential WBC count from non-breeding, healthy adult gannets are not available, it was not possible to determine with certainty if the higher occurrence of eosinophils was indicative of eosinophilia (an increase in the number of eosinophils), typically resulting from either a microbial/parasitic infection or an inflammatory response [3], or if this is the norm for gannets.

This study did not find any significant differences between sexes in any of the haematological parameters, which is in agreement with several previous studies conducted on a range of free-living seabird species ([17] on Adélie penguins *Pygoscelis adeliae*; [48] on northern fulmars *Fulmarus glacialis*; [47] on Cory's shearwaters *Calonectris diomedea*; [49] on red-tailed tropicbirds *Phaethon rubricauda*). This is, however, contradictory to other studies that found females ([16] on great tits *Parus major*; [50] on

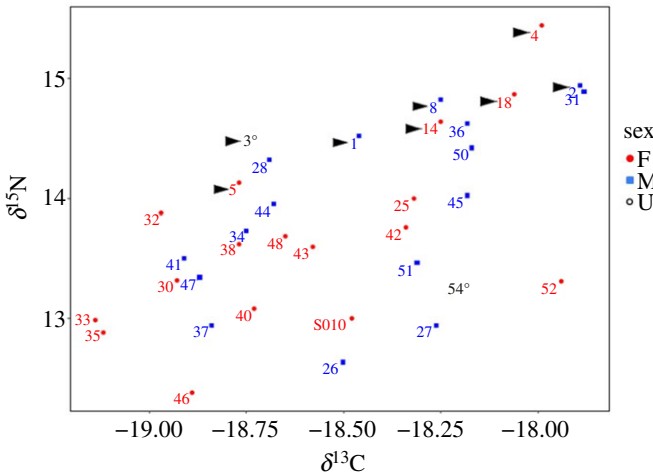

**Figure 2.** The results of stable isotope analysis from gannet blood. $\delta^{15}N$ values increase with each trophic level, $\delta^{13}C$ values decrease from coastal to offshore waters. F, female; M, male; U, sex not known. Individuals marked with arrowhead were sampled in 2017.

**Table 5.** Average values for foraging behaviours of male and female northern gannets. Values indicated as mean ± s.d. *Indicates significantly different values for the two sexes (see tables 7, 9 and 10 for details of statistical tests).

| parameter | year | female (mean ± s.d.) | male (mean ± s.d.) |
|---|---|---|---|
| average trip distance (km) | 2017 | 572.83 ± 481.73 | 600.65 ± 296.03 |
| | 2018* | 536.64 ± 143.01 | 368.64 ± 216.07 |
| furthest distance travelled (km) | 2017 | 208.41 ± 133.78 | 189.54 ± 70.38 |
| | 2018 | 255.43 ± 73.26 | 190.93 ± 85.77 |
| % time resting | 2017 | 37.44 ± 15.31 | 39.33 ± 9.39 |
| | 2018* | 46.34 ± 6.49 | 37.40 ± 7.87 |
| % time searching | 2017 | 38.11 ± 17.58 | 38.22 ± 9.63 |
| | 2018* | 29.10 ± 5.14 | 38.90 ± 6.60 |
| % time travelling | 2017 | 24.46 ± 8.89 | 22.46 ± 9.10 |
| | 2018 | 24.56 ± 5.24 | 23.70 ± 10.17 |

Magellanic penguins *Spheniscus magellanicus*) or males ([49] on brown boobies *Sula leucogaster*) to be in poorer health based on haematological parameters. An absence of sex-based differences in haematological parameters in the present study might indicate that reproduction, or at least chick provisioning, has similar physiological demands in both males and females. However, the time of sampling is likely to play a crucial role in the detection of intersexual differences. For example, if blood samples were collected soon after egg laying, females may show poorer condition since egg production can be physiologically expensive [51].

The ratio of heterophils to lymphocytes (H : L) was, to the best of our knowledge, the highest reported for any species under non-experimental conditions. This ratio is known to increase in response to stress in birds [23] and is considered to be a more reliable indicator of stress than plasma corticosteroid levels [15]. H : L ratios ranged from 2 to 9.1, slightly higher than reported by Moseley [27] in cape gannets (*Morus capensis*) (H : L ratio of 1.1–8.1), but substantially higher than reported by Brousseau-Fournier *et al.* [52] in non-breeding, adult northern gannets (H : L ratio of 0.18, the smallest to have been documented for any avian species). However, the high H : L ratio observed in this study is not entirely unexpected, since blood was sampled at change-over from the departing parent. These individuals would have been at the nest for a number of days waiting for partners to return from a foraging trip and experienced stress due to starvation. The H : L ratio is known to be elevated in birds that are

starved, infected, emaciated or injured [53]. Thus, the higher H : L ratios observed might also be indicative of an infection, in addition to starvation. Gross & Siegel [15] showed that the H : L ratio is also positively correlated to the magnitude of the stressor, and such a marked increase in H : L ratio might indicate that the breeding season is highly stressful for gannets.

A weak, but significant positive correlation between the total WBC count and the thrombocyte count was observed in this study. The WBC count represents immunocompetence, and leucocytosis (an increase in the number of leucocytes) is associated with infections (bacterial, viral or parasitic), inflammation, trauma, toxicity or haemorrhage. Though the primary role of avian thrombocytes is to facilitate the process of blood coagulation or haemostasis, they are also known to be involved in phagocytosis; however, their role in fighting infectious agents has not been properly established [3]. Thrombocytopenia (a decrease in the number of thrombocytes) and leucopenia (a decrease in the number of leucocytes) are both caused by excessive usage or depletion of said cells as in an infection, while thrombocytosis is usually indicative of a rebound response following haemorrhage or inflammation [5]. A correlation between WBC and thrombocyte count might reflect the involvement of thrombocytes in an immune response. However, it must be kept in mind that gannets are highly aggressive birds and jabbing, often violent in the colony, is very common [54]. Injuries sustained due to this behaviour may result in thrombocytosis or thrombocytopenia, and open wounds due to such injuries may result in inflammation and increased chances of infection, resulting in elevated or suppressed WBC counts.

There was a weak negative correlation between the eosinophil percentage and the percentage of time spent searching. Although the exact function of avian eosinophils has not yet been determined, it has been suggested that they might be involved in delayed hypersensitivity reactions and generalized inflammation [3], while previous studies on non-avian vertebrates have suggested that eosinophil numbers are related to parasite resistance [55,56]. If eosinophilia is indeed an indicator of parasitic load, then our results suggest that birds with parasitic infections spend less time searching for food. This may suggest a distinct link between haematological parameters and foraging behaviour that warrants further investigation.

It has been previously suggested that feeding at higher trophic levels by males might relate to a higher consumption of discards by this sex [34]. However, we detected no significant difference in the carbon or nitrogen stable isotope ratios between sexes consistent with sex differences in diet, nor any significant differences between the haematological parameters between sexes. We found no overall link between the haematological condition of gannets and their diet, consistent with a previous study on cape gannets (*M. capensis*). Moseley [27] studied two populations of cape gannets in South Africa, one that had been feeding on fishery discards for a considerable time period (2004–2008), and the other that fed predominantly on natural prey. In spite of this known difference in diet, there were no significant differences in the morphological indicators of body condition—body mass, pectoral muscle thickness and condition index (ratio of body mass to wing length)—between the two populations.

Of interest, gannet D27 (male) was feeding at a lower trophic level, showed abnormal blood counts and undertook exceptionally short foraging trips close to shore, suggesting a possible link between haemotological condition, diet and foraging behaviour. This gannet had the highest H : L ratio, indicating that it was highly stressed, and the high eosinophil % suggests that this stress was due to infection.

Immature erythrocyte percentage represents the regeneration of RBC, with higher regeneration possibly associated with anaemia. For most birds, the immature erythrocyte percentage was below the level consistent with anaemia, and in birds where it did exceed this limit, it was not remarkably elevated. Some sampled birds did show reduced or elevated numbers for the leucocyte differential counts; however, it has been established that the state of any one blood parameter does not indicate global immunocompetence or immunosuppression, as trade-offs are known to occur between the different types of leucocytes [57]. This also reinforces the fact that birds do not exhibit obvious signs until severely affected [21], and haematological studies can provide valuable insights into the bird's physiological status before symptoms appear, as in the case of gannet D27. Furthermore, the fact that sampled birds were breeding in itself indicates that they were not in very poor health, since seabirds are known to reduce their reproductive effort, to the extent of not even attempting to breed when in poor condition [27,58].

This study did not find a definitive link between haematological condition and diet in northern gannets. This is possibly due to a lack of severely immunocompromised or unhealthy birds sampled. Nevertheless, haematological analysis appears to be a more refined way of assessing health in birds [59]. Further work to define ranges for haematological parameters in pre- and post-breeding seabirds would provide baseline reference values for comparison and provide a better understanding of the physiological responses to stress at different stages of their reproductive cycle.

Ethics. This research was approved by the University College Cork Animal Ethics Committee, and all field procedures were conducted under licence by the National Parks & Wildlife Service, The British Trust for Ornithology and the Irish Health Products Regulatory Authority.

Data accessibility. All data associated with the study including raw tracking data, raw stable isotope analysis results and counts of identified cells are provided. Tracking data and SIA data have been deposited in the Dryad Digital Repository: https://doi.org/dryad.h70rxwdf6 [60].

Authors' contributions. M.J. and S.A.L. conceived the study. A.B., M.J. and Z.M. conducted the fieldwork including blood and tracking data collection. Z.M. performed haematological analysis under the guidance of S.A.L., and A.B. conducted analysis of tracking data. Z.M. led the writing of the manuscript with input from M.J. and S.A.L. All authors commented on the manuscript and gave final approval for publication.

Competing interests. The authors have no competing interests.

Funding. A.B. was funded by an Irish Research Council Postgraduate Scholarship (GOIPG 2016/503). Fieldwork and telemetry devices were funded through the Department of Agriculture, Fisheries, and the Marine FishKOSM project under grant (15/S/744).

Acknowledgements. We thank Sinead Morris for preliminary work on gannet haemotology that influenced this work as well as Sam Bayley, Tim McCarthy and Ailbhe Kavanagh for valuable assistance in the field. Two anonymous reviewers provided helpful comments to improve the manuscript.

# Appendix A

See tables 6–12.

**Table 6.** Sex-based differences in body weight, and haematological and stable isotope parameters for samples collected in 2018 ($n = 26$). Test statistic is '*t*' for independent samples *t*-test and '*W*' for Mann–Whitney *U*-test.

| parameter | test | test statistic | *p*-value |
|---|---|---|---|
| weight | Mann–Whitney *U*-test | 94.5 | 0.383 |
| immature erythrocyte % | *t*-test | 0.49 | 0.625 |
| microcyte % | Mann–Whitney *U*-test | 66 | 0.346 |
| total WBC | Mann–Whitney *U*-test | 70 | 0.472 |
| H : L ratio | Mann–Whitney *U*-test | 79.5 | 0.817 |
| heterophil % | *t*-test | −0.54 | 0.598 |
| lymphocyte % | *t*-test | 0.28 | 0.783 |
| eosinophil % | *t*-test | 0.47 | 0.641 |
| monocyte % | Mann–Whitney *U*-test | 75.5 | 0.639 |
| thrombocyte count | Mann–Whitney *U*-test | 83.5 | 0.979 |
| $\delta^{15}$N | *t*-test | −1.75 | 0.095 |
| $\delta^{13}$C | *t*-test | −1.52 | 0.142 |

**Table 7.** Sex-based differences in body weight and foraging parameters for birds sampled in 2017 ($n = 16$). Test statistic is '*t*' for independent samples *t*-test and '*W*' for Mann–Whitney *U*-test.

| parameter | test | test statistic | d.f. | *p*-value |
|---|---|---|---|---|
| weight | *t*-test | 1.49 | 14 | 0.157 |
| average trip distance (km) | Mann–Whitney *U*-test | 29 | — | 0.798 |
| furthest distance travelled (km) | Mann–Whitney *U*-test | 31 | — | 0.959 |
| % time resting | Mann–Whitney *U*-test | 30 | — | 0.878 |
| % time searching | Mann–Whitney *U*-test | 26 | — | 0.574 |
| % time travelling | *t*-test | 0.44 | 13.99 | 0.663 |

**Table 8.** Sex-based differences in haematological parameters for samples collected in 2017 ($n = 8$). Test statistic is '$t$' for independent samples $t$-test and '$W$' for Mann–Whitney $U$-test.

| parameter | test | test statistic | d.f. | $p$-value |
|---|---|---|---|---|
| immature erythrocyte % | $t$-test | 0.44 | 5.27 | 0.680 |
| microcyte % | $t$-test | −1.06 | 2.81 | 0.370 |
| thrombocyte count | Mann–Whitney $U$-test | 6 | — | 0.763 |

**Table 9.** Year-based differences between the haematological and stable isotope ($n = 16$) and foraging parameters ($n = 20$) for sampled males. Test statistic is '$t$' for independent samples $t$-test and '$W$' for Mann–Whitney $U$-test.

| parameter | test | test statistic | d.f. | $p$-value |
|---|---|---|---|---|
| weight | Mann–Whitney $U$-test | 15.5 | — | 0.636 |
| immature erythrocyte % | $t$-test | 0.37 | 2.86 | 0.738 |
| thrombocyte count | Mann–Whitney $U$-test | 10.5 | — | 0.252 |
| $\delta^{15}$N | $t$-test | 4.39 | 11.78 | <0.001 |
| $\delta^{13}$C | $t$-test | 1.46 | 3.38 | 0.230 |
| average trip distance (km) | $t$-test | 1.90 | 11.9 | 0.081 |
| furthest distance travelled (km) | $t$-test | −0.04 | 17.07 | 0.969 |
| % time resting | $t$-test | 0.48 | 13.25 | 0.639 |
| % time searching | $t$-test | −0.18 | 11.36 | 0.863 |
| % time travelling | Mann–Whitney $U$-test | 45 | — | 0.851 |

**Table 10.** Year-based differences between the haematological, stable isotope and foraging parameters ($n = 17$) for sampled females. Test statistic is '$t$' for independent samples $t$-test and '$W$' for Mann–Whitney $U$-test.

| parameter | test | test statistic | d.f. | $p$-value |
|---|---|---|---|---|
| weight | $t$-test | −1.07 | 6.95 | 0.320 |
| immature erythrocyte % | $t$-test | 0.70 | 5.09 | 0.513 |
| microcyte % | Mann–Whitney $U$-test | 50.5 | — | 0.077 |
| $\delta^{15}$N | $t$-test | 4.71 | 4.45 | 0.007 |
| $\delta^{13}$C | $t$-test | 2.06 | 4.95 | 0.095 |
| average trip distance (km) | Mann–Whitney $U$-test | 29 | — | 0.541 |
| furthest distance travelled (km) | $t$-test | −0.88 | 10.57 | 0.397 |
| % time resting | Mann–Whitney $U$-test | 23 | — | 0.236 |
| % time searching | Mann–Whitney $U$-test | 44 | — | 0.481 |
| % time travelling | $t$-test | −0.03 | 11.07 | 0.976 |

**Table 11.** Results of correlation analyses of haematological and stable isotope parameters for samples collected in 2018 ($n = 26$, d.f. = 24 for Pearson's product–moment correlation). Test statistic is '$r$' for Pearson's product–moment correlation and '$r_s$' for Spearman's rank correlation.

| parameter 1 | parameter 2 | test | test statistic | *p*-value |
|---|---|---|---|---|
| $\delta^{15}$N | immature erythrocyte % | Pearson | −0.12 | 0.570 |
| | microcyte % | Spearman | −0.29 | 0.152 |
| | total WBC | Spearman | 0.11 | 0.599 |
| | H : L ratio | Spearman | −0.19 | 0.354 |
| | heterophil % | Pearson | 0.03 | 0.888 |
| | lymphocyte % | Pearson | 0.20 | 0.317 |
| | eosinophil % | Pearson | −0.12 | 0.540 |
| | monocyte % | Spearman | −0.09 | 0.658 |
| | thrombocyte count | Spearman | 0.12 | 0.549 |
| $\delta^{13}$C | immature erythrocyte % | Pearson | 0.19 | 0.342 |
| | microcyte % | Spearman | <0.01 | 0.979 |
| | total WBC | Spearman | −0.14 | 0.483 |
| | H : L ratio | Spearman | 0.16 | 0.422 |
| | heterophil % | Pearson | 0.33 | 0.093 |
| | lymphocyte % | Pearson | −0.11 | 0.586 |
| | eosinophil % | Pearson | −0.26 | 0.194 |
| | monocyte % | Spearman | −0.03 | 0.872 |
| | thrombocyte count | Spearman | −0.35 | 0.082 |
| total WBC | immature erythrocyte % | Spearman | −0.08 | 0.709 |
| | microcyte % | Spearman | −0.17 | 0.402 |

**Table 12.** Results of correlation analyses of foraging and haematological parameters for samples collected in 2018 ($n = 20$, d.f. = 18 for Pearson's product–moment correlation). Test statistic is '$r$' for Pearson's product–moment correlation and '$r_s$' for Spearman's rank correlation.

| parameter 1 | parameter 2 | test | test statistic | *p*-value |
|---|---|---|---|---|
| average trip distance (km) | immature erythrocyte % | Pearson | 0.09 | 0.709 |
| | microcyte % | Spearman | 0.24 | 0.304 |
| | total WBC | Pearson | 0.02 | 0.926 |
| | H : L ratio | Spearman | −0.17 | 0.472 |
| | heterophil % | Pearson | −0.26 | 0.257 |
| | lymphocyte % | Pearson | 0.02 | 0.945 |
| | eosinophil % | Pearson | 0.26 | 0.268 |
| | monocyte % | Spearman | <0.01 | 0.983 |
| | thrombocyte count | Spearman | −0.13 | 0.584 |

(*Continued.*)

| parameter 1 | parameter 2 | test | test statistic | *p*-value |
|---|---|---|---|---|
| furthest distance travelled (km) | immature erythrocyte % | Pearson | −0.09 | 0.715 |
| | microcyte % | Spearman | 0.40 | 0.077 |
| | total WBC | Pearson | 0.05 | 0.828 |
| | H : L ratio | Spearman | −0.29 | 0.204 |
| | heterophil % | Pearson | −0.33 | 0.152 |
| | lymphocyte % | Pearson | 0.09 | 0.682 |
| | eosinophil % | Pearson | 0.28 | 0.222 |
| | monocyte % | Spearman | −0.02 | 0.925 |
| | thrombocyte count | Spearman | <0.01 | 0.995 |
| % time resting | immature erythrocyte % | Pearson | −0.08 | 0.727 |
| | microcyte % | Spearman | 0.02 | 0.941 |
| | total WBC | Pearson | 0.24 | 0.305 |
| | H : L ratio | Spearman | 0.09 | 0.679 |
| | heterophil % | Pearson | −0.14 | 0.568 |
| | lymphocyte % | Pearson | −0.19 | 0.424 |
| | eosinophil % | Pearson | 0.28 | 0.237 |
| | monocyte % | Spearman | 0.05 | 0.826 |
| | thrombocyte count | Spearman | 0.17 | 0.467 |
| % time searching | immature erythrocyte % | Pearson | −0.06 | 0.809 |
| | microcyte % | Spearman | −0.09 | 0.687 |
| | total WBC | Pearson | <0.01 | 0.975 |
| | H : L ratio | Spearman | 0.05 | 0.830 |
| | heterophil % | Pearson | 0.34 | 0.119 |
| | lymphocyte % | Pearson | 0.18 | 0.442 |
| | monocyte % | Spearman | 0.02 | 0.945 |
| | thrombocyte count | Spearman | −0.08 | 0.721 |
| % time travelling | immature erythrocyte % | Pearson | 0.14 | 0.559 |
| | microcyte % | Spearman | 0.04 | 0.859 |
| | total WBC | Pearson | −0.25 | 0.279 |
| | H : L ratio | Spearman | −0.11 | 0.638 |
| | heterophil % | Pearson | −0.19 | 0.409 |
| | lymphocyte % | Pearson | 0.02 | 0.919 |
| | eosinophil % | Pearson | 0.22 | 0.354 |
| | monocyte % | Spearman | −0.21 | 0.368 |
| | thrombocyte count | Spearman | −0.08 | 0.719 |

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
