## [Reviewer comments · Royal Society Open Science]

Review History

RSOS-192164.R0 (Original submission)

Review form: Reviewer 1

Is the manuscript scientifically sound in its present form?

Yes

Are the interpretations and conclusions justified by the results?

Yes

Is the language acceptable?

Yes

Do you have any ethical concerns with this paper?

No

Have you any concerns about statistical analyses in this paper?

No

Recommendation?

Accept with minor revision (please list in comments)

Comments to the Author(s)

I think in general this is a very useful and interesting paper and the authors should be pleased with the quality of paper they have produced. I think more detail is needed in the methods section and a small number of other changes would help with clarity and ease of reading in certain sections, but overall this is a good paper to read.

Review form: Reviewer 2**Is the manuscript scientifically sound in its present form?**

Yes

Are the interpretations and conclusions justified by the results?

No

Is the language acceptable?

Yes

Do you have any ethical concerns with this paper?

No

Have you any concerns about statistical analyses in this paper?

No

Recommendation?

Accept with minor revision (please list in comments)

Comments to the Author(s)

Dear Authors,

This was a nice paper to read and I laud the efforts in your study to tie these factors together in a species. Mostly I think you have succeeded, although as I note below, I really feel that you overstate the implications from that 1 bird (D27). But otherwise, my comments are very minor and I suggest would take little time to fix up. I appreciated that you have moved this field forward with this work.

P 4, L 17-22: It was a bit surprising not to see this recent paper cited, which presented leukocyte data and H:L ratios for a variety of seabirds and discussed issues of stress and migration: Mallory, M.L., Little, C.M., Boyd, E.S., Ballard, J., Elliott, K.H., Gilchrist, H.G., Hipfner, J.M., Petersen, A. and Shutler, D., 2015. Leucocyte profiles of Arctic marine birds: correlates of migration and breeding phenology. *Conservation physiology*, 3(1), p.cov028.

P 6, L 14: Why is "Northern" capitalized but not "gannet". I do not think I would capitalize either

P 16, L 26-28: I am dubious about a statement where you linked foraging metrics and blood metrics in a single bird. You simply cannot be sure that there is an associative linkage there.

P16, L38-42; P17 L18-20: This seems like another location where a comparison to the seabird species presented in Mallory et al (2015) seems apt.

P18 L40-48: Your data do not show any meaningful difference in nitrogen isotopes and you acknowledge this, and then you turn around and say that there are sex-based differences in diet. How can that be - your data definitely do not show that. Suggest this entire paragraph be tossed.

P19 L3-16: I do not think this adds anything to your manuscript. Recommend removing.

P19 L18-45: This is an awful lot of speculation for a single bird, which, for all you know, might have had a disease affecting it (as you suggest). The implication you leave is that there was a "link" between foraging range and trophic position etc., but you have very little basis for this, at least for sample size. I would reduce to 1 or 2 lines to note that it was an oddity and with additional data might point to linkages between those parameters.

P20 L16-18: Each time you refer to this gannet you keep stating it more strongly; here it says "confirms"! I do not think a single observation from a correlational study confirms anything - please rephrase.

Decision letter (RSOS-192164.R0)

24-Feb-2020

Dear Dr Jessopp,

The editors assigned to your paper ("Evidence of links between haematologic condition and foraging behaviour in Northern gannets (*Morus bassanus*)") have now received comments from reviewers. We would like you to revise your paper in accordance with the referee and Associate Editor suggestions which can be found below (not including confidential reports to the Editor). Please note this decision does not guarantee eventual acceptance.

Please submit a copy of your revised paper before 18-Mar-2020. Please note that the revision deadline will expire at 00.00am on this date. If we do not hear from you within this time then it will be assumed that the paper has been withdrawn. In exceptional circumstances, extensions may be possible if agreed with the Editorial Office in advance. We do not allow multiple rounds of revision so we urge you to make every effort to fully address all of the comments at this stage. If deemed necessary by the Editors, your manuscript will be sent back to one or more of the original reviewers for assessment. If the original reviewers are not available, we may invite new reviewers.

If your study uses humans or animals please include details of the ethical approval received, including the name of the committee that granted approval. For human studies please also detail

whether informed consent was obtained. For field studies on animals please include details of all permissions, licences and/or approvals granted to carry out the fieldwork.

- Data accessibility

If you wish to submit your supporting data or code to Dryad (<http://datadryad.org/>), or modify your current submission to dryad, please use the following link:
<http://datadryad.org/submit?journalID=RSOS&manu=RSOS-192164>

- Competing interests

- Authors' contributions

- Acknowledgements

- Funding statement

Kind regards,

Anita Kristiansen

Editorial Coordinator

on behalf of Dr Sean Rands (Associate Editor) and Kevin Padian (Subject Editor)
 openscience@royalsociety.org

Associate Editor's comments (Dr Sean Rands):

Comments to the Author:

Two reviewers have kindly commented on your manuscript, and I would like you to respond to their comments in a revision. Here are some additional comments from the reviewers that I am not sure are visible to you, and should be responded to:

"I think this is overall an interesting and well written paper.

I am a little surprised that references do not include either of the Campbell, or Campbell and Ellis, Avian hematology or Avian and exotic animal hematology and cytology textbooks as I would think those would be essential reading for this type of paper but perhaps the authors did not find that to be the case?"

"In a number of places in the text the negative correlation between time spent actively foraging and percentage of eosinophils is described as "weakly correlated" and other times it is just described as "correlated", so it would help to make this consistent throughout."

"I think more details needs to be included in the methods section such as % of body weight taken as blood sample, which feathers the GPS tracker was attached to, width of tape strips etc. Also explanation of why EDTA was used rather than just making smears of undiluted sample. In the staining step, step 2 has a wide range of seconds so this should be narrowed if possible for ease of replication.

Table 1 is somewhat confusing - it would be better to have separate columns for reference ranges from previous papers and those suggested by this study.

Table 2 also need to have units on the columns marking numbers of cells rather than just No. - as is shown in Table 3."

In addition, I have some requests, mainly focussed on properly presenting the results of your analyses (both significant and non-significant) in a form that will be of use to anyone wishing to use your results in a metaanalysis.

p7 l46: I'm not sure you used RStudio to conduct your analysis - it is an editor that you can run R within, and it's R that your using for the analysis. I'd stick with the citation of R instead, and dump the RStudio reference.

p8 l10: please give the full statistics here, rather than just the p values. Why run t tests *and* Mann_Whitney - I assume there's some issue of non-normal data, maybe?

p12 l14 etc.: please give full stats here too.

p13 l3: please give non-sig stats too - it might be better to present the details of table 4 in full.

p15 top: give the df for the Pearson, and the sample size for the spearman, and give the non-sig stats in full.

Please sort out the legend for figure 3. It's good that you have different symbols for the F/M/U data, but the legend currently has these symbols obscured by letters, which makes them less useful for black+white printing or colourblind readers.

Comments to Author:

Reviewers' Comments to Author:

Reviewer: 1

Comments to the Author(s)

I think in general this is a very useful and interesting paper and the authors should be pleased with the quality of paper they have produced. I think more detail is needed in the methods section and a small number of other changes would help with clarity and ease of reading in certain sections, but overall this is a good paper to read.

Reviewer: 2

Comments to the Author(s)

Dear Authors,

This was a nice paper to read and I laud the efforts in your study to tie these factors together in a species. Mostly I think you have succeeded, although as I note below, I really feel that you overstate the implications from that 1 bird (D27). But otherwise, my comments are very minor and I suggest would take little time to fix up. I appreciated that you have moved this field forward with this work.

P 4, L 17-22: It was a bit surprising not to see this recent paper cited, which presented leukocyte data and H:L ratios for a variety of seabirds and discussed issues of stress and migration: Mallory, M.L., Little, C.M., Boyd, E.S., Ballard, J., Elliott, K.H., Gilchrist, H.G., Hipfner, J.M., Petersen, A. and Shutler, D., 2015. Leucocyte profiles of Arctic marine birds: correlates of migration and breeding phenology. *Conservation physiology*, 3(1), p.cov028.

P 6, L 14: Why is "Northern" capitalized but not "gannet". I do not think I would capitalize either

P 16, L 26-28: I am dubious about a statement where you linked foraging metrics and blood metrics in a single bird. You simply cannot be sure that there is an associative linkage there.

P16, L38-42; P17 L18-20: This seems like another location where a comparison to the seabird species presented in Mallory et al (2015) seems apt.

P18 L40-48: Your data do not show any meaningful difference in nitrogen isotopes and you acknowledge this, and then you turn around and say that there are sex-based differences in diet. How can that be - your data definitely do not show that. Suggest this entire paragraph be tossed.

P19 L3-16: I do not think this adds anything to your manuscript. Recommend removing.

P19 L18-45: This is an awful lot of speculation for a single bird, which, for all you know, might have had a disease affecting it (as you suggest). The implication you leave is that there was a "link" between foraging range and trophic position etc., but you have very little basis for this, at least for sample size. I would reduce to 1 or 2 lines to note that it was an oddity and with additional data might point to linkages between those parameters.

P20 L16-18: Each time you refer to this gannet you keep stating it more strongly; here it says "confirms"! I do not think a single observation from a correlational study confirms anything - please rephrase.

Author's Response to Decision Letter for (RSOS-192164.R0)

See Appendix A.

Decision letter (RSOS-192164.R1)

06-Apr-2020

Dear Dr Jessopp,

It is a pleasure to accept your manuscript entitled "Evidence of links between haematologic condition and foraging behaviour in Northern gannets (*Morus bassanus*)" in its current form for publication in Royal Society Open Science. The comments of the Editors who reviewed your manuscript are included at the foot of this letter.

Kind regards,
Lianne Parkhouse
Editorial Coordinator
Royal Society Open Science
openscience@royalsociety.org

on behalf of Dr Sean Rands (Associate Editor) and Kevin Padian (Subject Editor)
openscience@royalsociety.org

Associate Editor Comments to Author (Dr Sean Rands):

Thank you for your revision, and apologies for the currently inevitable delay in responding (I hope you are all keeping well and safe). I'm happy with the work you have done with the revision, and have no further requirements from you.

Appendix A

Dear Dr Rands,

Thank you for taking the time to have our manuscript reviewed for *Royal Society Open Science*. We were particularly pleased by the positive response to our work and have addressed both your and the reviewers' comments as outlined in *Italics* below each point. We have also added sections on Ethics, data accessibility, author contributions, and competing interests to the manuscript. We hope that these changes meet with your approval and you consider the paper worthy of publishing in its revised format.

Associate Editor Comments:

I think this is overall an interesting and well written paper. I am a little surprised that references do not include either of the Campbell, or Campbell and Ellis, Avian hematology or Avian and exotic animal hematology and cytology textbooks as I would think those would be essential reading for this type of paper but perhaps the authors did not find that to be the case?"

Response

Unfortunately, neither of these were available within our institutional library system, but we have now purchased a copy of the most recent edition of Campbell (2015), and refer to this as the most up to date information available. Cross-checking this with the earlier Campbell 1995 shows that the identification of cell types is the same across texts.

In a number of places in the text the negative correlation between time spent actively foraging and percentage of eosinophils is described as "weakly correlated" and other times it is just described as "correlated", so it would help to make this consistent throughout.

Response

This has been made consistent as "weakly negatively correlated" so as not to overstate the strength of this relationship.

I think more details needs to be included in the methods section such as % of body weight taken as blood sample, which feathers the GPS tracker was attached to, width of tape strips etc. Also explanation of why EDTA was used rather than just making smears of undiluted sample. In the staining step, step 2 has a wide range of seconds so this should be narrowed if possible for ease of replication.

Response

We have added details of the volume of blood taken with respect to body weight (~0.07% volume/weight) as well as details of the feathers that tags were attached to (2-3 central tail feathers) and details of tape (3-4 strips of Tesa waterproof tape, 15mm width). We have also provided text explaining that EDTA is used as an anti-coagulant and preservative to fix cells prior to smearing, referencing Campbell (2015). The EDTA facilitates the best preservation of cellular components, morphology of blood cells, and ensures structural integrity of the cells during the physical process of smearing thus avoiding 'crush artifacts'. The times reported for step 2 represent those in the protocol for the staining kit. We have however modified this based on the actual times used for this study (~45 seconds, p 6 l16).

Table 1 is somewhat confusing - it would be better to have separate columns for reference ranges from previous papers and those suggested by this study.

Response

We have amended the table to present this information more clearly, but have avoided separating this into two smaller tables as the information is a compilation of reference values for this species across a range of studies, including ours.

Table 2 also need to have units on the columns marking numbers of cells rather than just No. - as is shown in Table 3.

Response

We have removed the column for No. of each cell type as this is simply calculated from the % of each cell type from the total WBC count and does not provide any additional information, particularly as reference values are based on %.

In addition, I have some requests, mainly focussed on properly presenting the results of your analyses (both significant and non-significant) in a form that will be of use to anyone wishing to use your results in a metaanalysis.

Response

We have now provided more details on the test values and significance levels for all tests, regardless of significance. Because of the large number of tests, we have provided these details as an appendix so as not to detract from the flow of the text.

p7 l46: I'm not sure you used RStudio to conduct your analysis - it is an editor that you can run R within, and it's R that your using for the analysis. I'd stick with the citation of R instead, and dump the RStudio reference.

Response

We have amended the text to state that analysis was performed in R and changed the citation accordingly.

p8 l10: please give the full statistics here, rather than just the p values. Why run t tests *and* Mann_Whitney - I assume there's some issue of non-normal data, maybe?

Response

Indeed this is because some variables were normally distributed and therefore parametric test was appropriate, while other variables were not and so non-parametric tests were performed. We have amended the text to make this clearer as follows "Haematological and behavioural variables were tested for normality, and differences between sex and year tested using either parametric independent-samples t-test or non-parametric Mann-Whitney U-test accordingly."

p12 l14 etc.: please give full stats here too.

Response

Amended as suggested

p13 l3: please give non-sig stats too - it might be better to present the details of table 4 in full.

Response

Amended as suggested

p15 top: give the df for the Pearson, and the sample size for the spearman, and give the non-sig stats in full.

Response

Amended as suggested

Please sort out the legend for figure 3. It's good that you have different symbols for the F/M/U data, but the legend currently has these symbols obscured by letters, which makes them less useful for black+white printing or colourblind readers.

Response

The legend has now been fixed and is more clearly visible.

Reviewers' Comments to Author:

Reviewer: 1

I think in general this is a very useful and interesting paper and the authors should be pleased with the quality of paper they have produced. I think more detail is needed in the methods section and a small number of other changes would help with clarity and ease of reading in certain sections, but overall this is a good paper to read.

Response

We would like to thank the reviewer for this very positive review. We have added additional detail to the methods which also address the comments of other reviewers, which are outlined in more detail within responses to comments from the Associate Editor and Reviewer 2

Reviewer: 2

This was a nice paper to read and I laud the efforts in your study to tie these factors together in a species. Mostly I think you have succeeded, although as I note below, I really feel that you overstate the implications from that 1 bird (D27). But otherwise, my comments are very minor and I suggest would take little time to fix up. I appreciated that you have moved this field forward with this work.

P 4, L 17-22: It was a bit surprising not to see this recent paper cited, which presented leukocyte data and H:L ratios for a variety of seabirds and discussed issues of stress and migration:

Mallory, M.L., Little, C.M., Boyd, E.S., Ballard, J., Elliott, K.H., Gilchrist, H.G., Hipfner, J.M., Petersen, A. and Shutler, D., 2015. Leucocyte profiles of Arctic marine birds: correlates of migration and breeding phenology. *Conservation physiology*, 3(1), p.cov028.

Response

We thank the reviewer for pointing us in the direction of this paper which we have now included. This has now been referred to on pages 4 and 5 of the introduction.

P 6, L 14: Why is "Northern" capitalized but not "gannet". I do not think I would capitalize either

Response

*Since many birds have descriptive common names, naming nomenclature dictates that the first part of the common name is capitalised to differentiate the name from the description. 'Northern gannet' refers to the name of the bird rather than 'gannet' that occurs in the north. We have adjusted the text and referred to "Northern gannet (*Morus bassanus*), hereafter gannet" at first mention of the species, and then referred to "gannet" at all subsequent mentions.*

P 16, L 26-28: I am dubious about a statement where you linked foraging metrics and blood metrics in a single bird. You simply cannot be sure that there is an associative linkage there.

Response

This statement has been removed, and discussion of this individual further down has been significantly reduced as suggested below.

P16, L38-42; P17 L18-20: This seems like another location where a comparison to the seabird species presented in Mallory et al (2015) seems apt.

Response

The Mallory et al. (2015) paper indeed provides some valuable reference ranges for several marine birds. However, it would not be very appropriate to compare these directly with our observations here because these species are not very closely related to gannets, and the blood samples in Mallory et al. (2015) were collected during incubation, while those in our study were collected during chick-rearing. However, we have referred to H:L ratios found in a different study on cape gannets, which we think is a better comparison due the species being more closely related, having more similar biology, and the blood samples being collected at a similar time during the breeding cycle. Instead we refer to this paper here in terms of the H:L ratio increasing as a response to stress.

P18 L40-48: Your data do not show any meaningful difference in nitrogen isotopes and you acknowledge this, and then you turn around and say that there are sex-based differences in diet. How can that be – your data definitely do not show that. Suggest this entire paragraph be tossed.

Response

This paragraph has been replaced with a statement that we found no sex based differences in diet or hematological parameters, and no link between diet and blood parameters.

P19 L3-16: I do not think this adds anything to your manuscript. Recommend removing.

Response

This paragraph has been heavily edited down to simply state that the weak link between eosinophils and time spent foraging may be related to parasite load, and that this warrants further investigation

P19 L18-45: This is an awful lot of speculation for a single bird, which, for all you know, might have had a disease affecting it (as you suggest). The implication you leave is that there was a “link” between foraging range and trophic position etc., but you have very little basis for this, at least for sample size. I would reduce to 1 or 2 lines to note that it was an oddity and with additional data might point to linkages between those parameters.

Response

We agree with the reviewer and detailed text explaining the possible causes of the high eosinophil count and HL ratio has been removed, with the observation limited to 2 sentences.

P20 L16-18: Each time you refer to this gannet you keep stating it more strongly; here it says “confirms”! I do not think a single observation from a correlational study confirms anything – please rephrase.

Response

The reviewer is quite correct and we have removed mention of this bird here. Instead we suggest that haematological analysis is likely to be a more refined method of assessing bird health than diet or foraging metrics.